# SARS-CoV-2 S protein:ACE2 interaction reveals novel allosteric targets

Palur V Raghuvamsi[1,2†], Nikhil K Tulsian[1,3†], Firdaus Samsudin[2], Xinlei Qian[4], Kiren Purushotorman[4], Gu Yue[4], Mary M Kozma[4], Wong Y Hwa[5], Julien Lescar[5], Peter J Bond[1,2*], Paul A MacAry[4*], Ganesh S Anand[1,6*]

[1]Department of Biological Sciences, National University of Singapore, Singapore, Singapore; [2]Bioinformatics Institute, Agency for Science, Technology, and Research (A*STAR), Singapore, Singapore; [3]Centre for Life Sciences, Department of Biochemistry, National University of Singapore, Singapore, Singapore; [4]Life Sciences Institute, Centre for Life Sciences, National University of Singapore, Singapore, Singapore; [5]School of Biological Sciences, Nanyang Technological University, Singapore, Singapore; [6]Current address: Department of Chemistry, Department of Biochemistry and Molecular Biology, Center for Infectious Disease Dynamics -Huck Institute of the Life Sciences, The Pennsylvania State University, University Park, United States

**Abstract** The spike (S) protein is the main handle for SARS-CoV-2 to enter host cells via surface angiotensin-converting enzyme 2 (ACE2) receptors. How ACE2 binding activates proteolysis of S protein is unknown. Here, using amide hydrogen–deuterium exchange mass spectrometry and molecular dynamics simulations, we have mapped the S:ACE2 interaction interface and uncovered long-range allosteric propagation of ACE2 binding to sites necessary for host-mediated proteolysis of S protein, critical for viral host entry. Unexpectedly, ACE2 binding enhances dynamics at a distal S1/S2 cleavage site and flanking protease docking site ~27 Å away while dampening dynamics of the stalk hinge (central helix and heptad repeat [HR]) regions ~130 Å away. This highlights that the stalk and proteolysis sites of the S protein are dynamic hotspots in the prefusion state. Our findings provide a dynamics map of the S:ACE2 interface in solution and also offer mechanistic insights into how ACE2 binding is allosterically coupled to distal proteolytic processing sites and viral–host membrane fusion. Thus, protease docking sites flanking the S1/S2 cleavage site represent alternate allosteric hotspot targets for potential therapeutic development.

**\*For correspondence:**
peterjb@bii.a-star.edu.sg (PJB);
micpam@nus.edu.sg (PAMA);
gsa5089@psu.edu (GSA)

[†]These authors contributed equally to this work

**Competing interests:** The authors declare that no competing interests exist.

## Introduction

The COVID-19 pandemic caused by the SARS-CoV-2 virus has sparked extensive efforts to map molecular details of its life cycle to drive vaccine and therapeutic discovery (*Bar-Zeev and Inglesby, 2020*). SARS-CoV-2 belongs to the family of *Coronaviridae,* which includes other human pathogens including common cold-causing viruses (hCoV-OC43, HKU, and 229E), SARS, and MERS-CoV (*Corman et al., 2018*; *St-Jean et al., 2004*; *Lau et al., 2006*; *Warnes et al., 2015*). SARS-CoV-2 has an ~30 kb RNA (positive stranded) genome with 14 open reading frames, encoding four structural proteins – spike (S) protein, membrane (M) protein, envelope (E) protein, and nucleo-protein; 16 non-structural proteins, and 9 accessory proteins (*Su et al., 2016*; *Andersen et al., 2020*; *Tan et al., 2005*) An intact SARS-CoV-2 virion consists of a nucleocapsid core containing genomic RNA within a lipid–protein envelope forming a spherical structure of diameter ~100 nm (*Ke et al., 2020*). The viral envelope is decorated with S, M, and E proteins (*Ke et al., 2020*). The prefusion S

protein is a club-shaped homotrimeric class I viral fusion protein that has distinctive 'head' and 'stalk' regions (*Figure 1A*).

A characteristic feature of SARS-CoV-2 is that upon host entry, its prefusion S protein is proteolyzed by host proteases into constituent S1 and S2 subunits. The S1 subunit comprises an N-terminal domain (NTD) and a receptor binding domain (RBD) that interacts with the host receptor angiotensin-converting enzyme-2 (ACE2) (*Lan et al., 2020*; *Hoffmann et al., 2020*) to initiate viral entry into the target cell (*Yan et al., 2020*). The defining virus–host interaction for entry is therefore that mediated by the viral S protein with the host ACE2 receptor (*Lan et al., 2020*). Binding to ACE2 primes the S protein for proteolysis by host furin proteases at the S1/S2 cleavage site (*Walls et al., 2017*; *Vankadari, 2020*). The S2 subunit consists of six constituent domains harboring the membrane fusion machinery of the virus. These comprise the fusion peptide (FP), heptad repeat (HR1), central helix (CH), heptad repeat 2 (HR2), connector domain (CD), transmembrane domain (TM), and cytoplasmic tail (CT) domain (*Walls et al., 2020*; *Wrapp et al., 2020*). Extensive structural studies (*Ke et al., 2020*; *Walls et al., 2020*; *Fan et al., 2020*; *Turoňová et al., 2020*) have captured S protein of coronaviruses in distinct open (PDB: 6VXX) (*Walls et al., 2020*) and closed (PDB: 6VYB) (*Walls et al., 2020*) conformational states relative to the orientation of the RBD. These structures additionally reveal distinct orientations of the ectodomain (ECD) in pre- and postfusion states and highlight the intrinsic ensemble nature of the S protein in solution. The S2 subunit promotes host–viral membrane fusion and viral entry (*Figure 1B*).

Despite extensive cryo-Electron Microscopy (cryo-EM) studies, a map of the S:ACE2 interface in solution and how ACE2 binding to the RBD primes enhanced proteolytic processing at the S1/S2 site is entirely unknown. Amide hydrogen/deuterium exchange mass spectrometry (HDXMS), together with molecular dynamics (MD) simulations, offers a powerful combined approach for

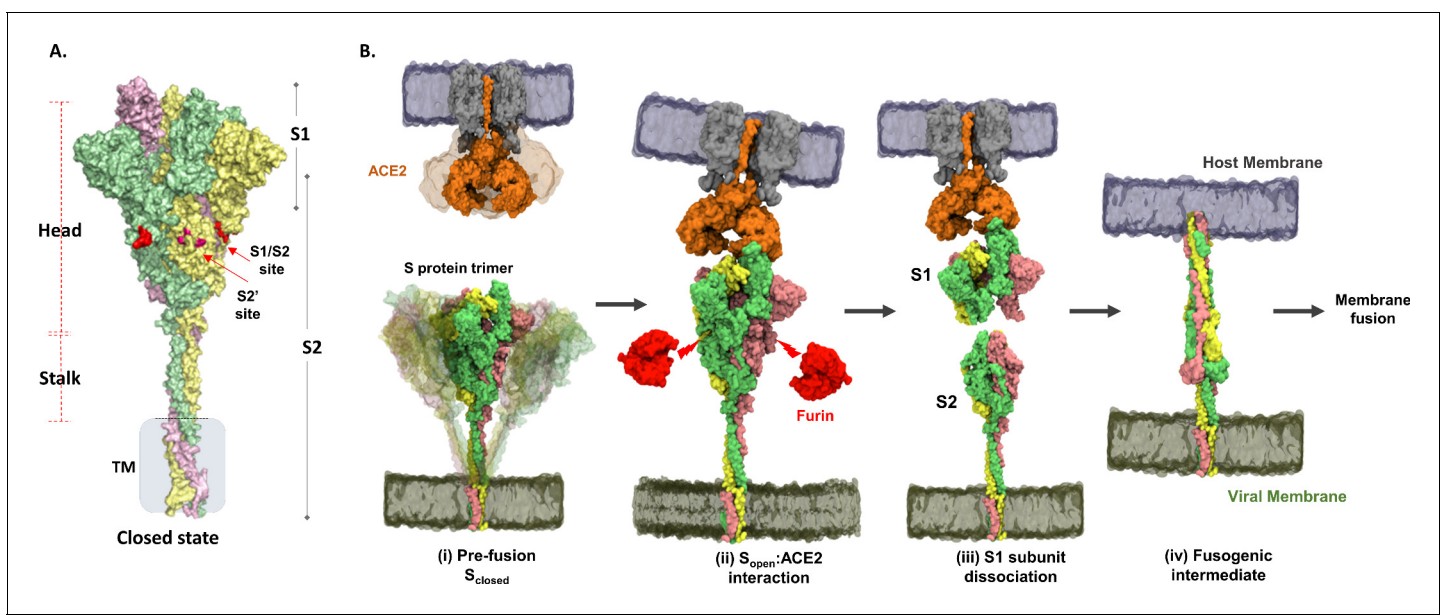

**Figure 1.** Structure and domain organization of trimeric spike (S) protein showing steps in the virus–host entry initiated by S recognition and binding to angiotensin-converting enzyme 2 (ACE2) receptor. (**A**) Prefusion S protein trimer in closed conformational state, with monomers shown in yellow, green, and pink. S protein construct (1–1245) used in this study showing head, stalk, and transmembrane (TM) segments as generated by integrative modeling. The S1/S2 and S2' cleavage sites are in red. Proteolytic processing (furin) of S protein generates S1 and S2 subunits. (**B**) Schematic of viral entry into host cell mediated by S:ACE2 interactions as previously outlined (*Shang et al., 2020*): (i) Intrinsic dynamics of prefusion S protein trimer decorating SARS-CoV-2 and host ACE2 dimeric structure showing sweeping motions of S protein and ACE2 to facilitate S:ACE2 recognition. (ii) In the open conformation (S$_{open}$), receptor binding domain adopts an 'up' orientation to recognize and bind the host membrane-bound ACE2 receptor (Protein Data Bank [PDB] ID: 1R42). ACE2 binding induces conformational changes promoting Furin* (red) proteolysis at the S1/S2 cleavage site (red arrows), leading to dissociation of S1 and S2 subunits, the mechanism of which is unknown. *Furin here also denotes relevant related proteases. (iii) The residual ACE2-bound S1 subunit becomes stably bound to ACE2 and S2 subunits dissociate. (iv) Conformational changes in the separated S2 subunit promote formation of an extended helical fusogenic intermediate (PDB ID: 6M3W) (*Fan et al., 2020*) for fusion into the host cell membrane, membrane fusion, and viral entry into the host cell (*Hoffmann et al., 2020*).

describing virus protein conformational dynamics and breathing (*Lim et al., 2017a*) and mapping protein–protein interactions for host receptor–virus interactions (*Lim et al., 2017b*). Here, we describe dynamics of free S protein and S:ACE2 complex, which reveal allosteric effects of ACE2 binding-induced conformational changes at distal stalk and protease docking sites flanking the S1/S2 cleavage sites. Our studies uncover distal 'hotspots' critical for the first step of the SARS-CoV-2 infection and thereby represent novel targets beyond the RBD for therapeutic intervention.

## Results and discussion

### Subunit-specific dynamics and domain motions of S protein trimer

Structural snapshots of the ACE2 binding interface with the SARS-CoV-2 S protein have previously been described for the RBD alone (*Lan et al., 2020*; *Wrapp et al., 2020*; *Ali and Vijayan, 2020*; *Chan et al., 2020*; *Wang et al., 2020a*). In this study, we have expanded this to map interactions and dynamics of ACE2 binding with a larger S protein construct, S (1–1208), lacking only the C-terminal membrane spanning helices. Mutations at the S1/S2 cleavage site (PRRAS motif substituted by PGSAS motif) and 986–987 (KV substituted PP) were engineered (*Wrapp et al., 2020*) to block host cell-mediated S protein proteolysis during expression and purification (*Figure 2—figure supplement 1*). S (1–1208), ACE2, and RBD eluted as trimers, dimers, and monomers, respectively, on size-exclusion chromatography (*Figure 2—figure supplement 1*, *Figure 3—figure supplement 1*, and *Figure 5—figure supplement 1*). S protein hereafter in the text denotes S (1–1208). Isolated RBD constructs showed high-affinity binding to ACE2 (*Figure 3—figure supplement 1*, *Figure 5—figure supplement 1*).

HDXMS of S protein alone was next carried out as described in 'Materials and methods'. Pepsin proteolysis of the S protein generated 317 peptides with high signal to noise ratios, yielding a primary sequence coverage of ~87% (*Figure 2—figure supplement 2*). S protein is highly glycosylated (at least 22 sites have been predicted and characterized on S protein) (*Watanabe et al., 2020*). Of these, 20 sites are predicted to be N-linked glycosylation modifications. We obtained peptides spanning 12 of the 20 predicted glycosylation sites. None of these peptides were glycosylated, making deuterium exchange of non-glycosylated peptides the focus of this study.

HDXMS results were overlaid onto integrative models of the full-length S protein trimer built using experimental structures of prefusion S ECD in the open conformation (PDB ID: 6VSB) (*Wrapp et al., 2020*) and HR2 domain from SARS S protein as templates. A deuterium exchange heat map (t = 1 and 10 min) revealed the stalk region to show the greatest relative deuterium exchange (*Figure 2A*). This is consistent with earlier studies showing at least 60° sweeping motions of the three identified hinge regions of the stalk (*Turoňová et al., 2020*). This was further verified via all-atom MD simulations of the S protein model embedded in a viral model membrane, which showed significant motions of the S protein ECD resulting from the high flexibility of the stalk region (*Figure 2B*), combined with large atomic fluctuations around the HR2 domain, compared to the rest of the protein (*Figure 2—figure supplement 3*, *Figure 2—figure supplement 4*).

Interestingly, the deuterium exchange heat map also showed highest relative exchange in the S2 subunit (*Figure 2—figure supplement 3*) and helical segments of the stalk, while peptides spanning the FP showed relatively lower deuterium exchange overall. Individually, S1 and S2 subunits showed different intrinsic deuterium exchange kinetics, where average relative fractional deuterium uptake (RFU) at early deuterium exchange time points of S1 subunit (~0.25) was lower than the average RFU (~0.35) for the S2 subunit (*Figure 2—figure supplement 3*, source data – *Figure 2—source data 1*). Furthermore, peptides connecting the RBD to the rest of the S protein showed greater deuterium exchange, suggesting a 'hinge' role for this segment to facilitate RBD adopting an ensemble of open and closed conformational states (*Figure 2C*). Indeed, in our simulations of the S protein (*Figure 2B*), the RBD oriented initially in an 'up' conformation and exhibited spontaneous motion toward the 'down' conformation relative to the hinge region (*Figure 2D*, *Figure 2—figure supplement 4A*). Interestingly, a part of the receptor binding motif, specifically residues 476–486, exhibited a higher degree of flexibility based on its average atomic fluctuations (*Figures 2B and 3B*), suggesting a role for the ACE2 receptor in stabilizing S protein dynamics and priming it for host furin proteolysis.

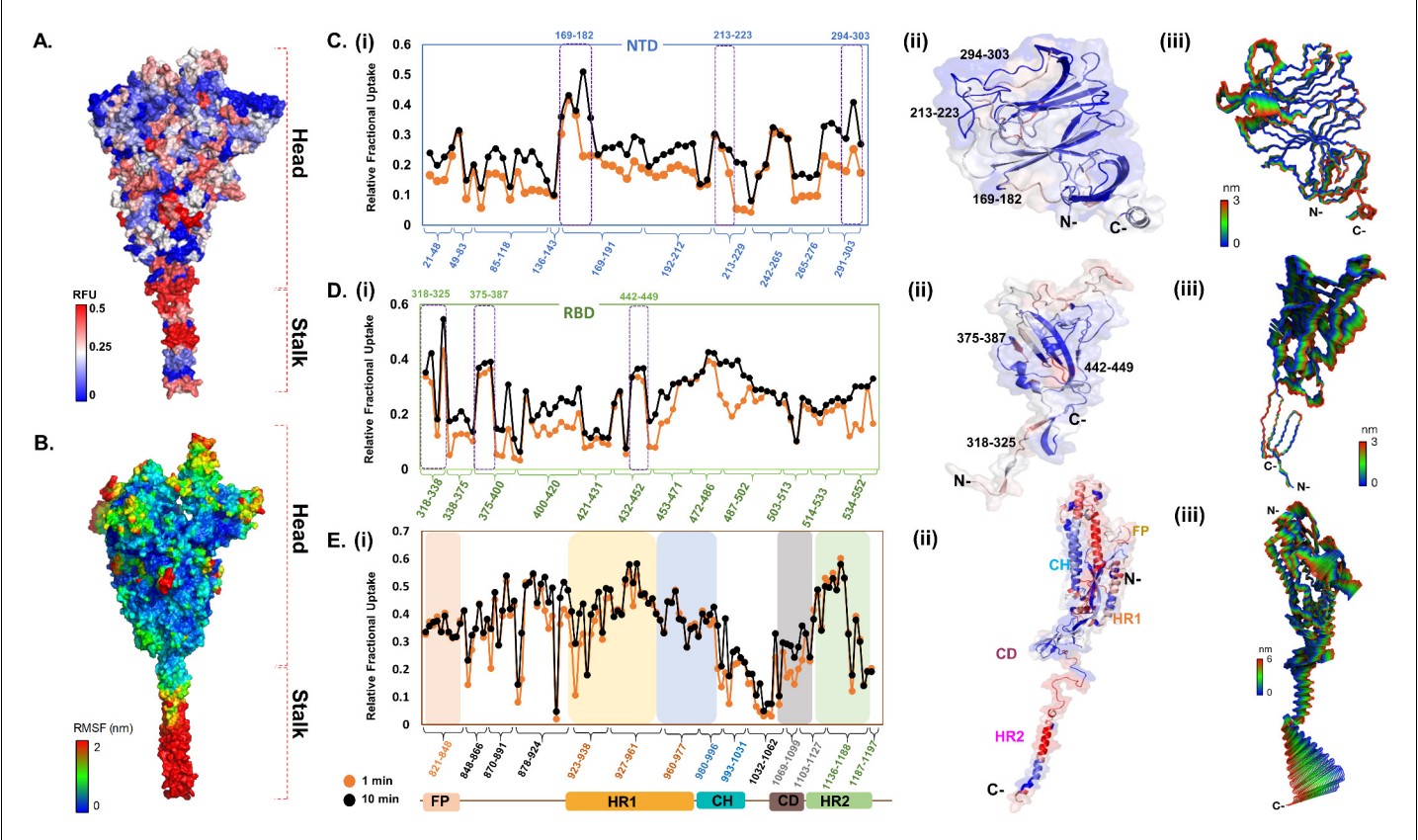

**Figure 2.** Deuterium exchange heat map and molecular dynamics simulations reveal domain-specific conformational dynamics of prefusion spike (S) protein trimer. (A) Deuterium exchange at t = 1 min deuterium exchange mapped onto the structure of S protein (shades of blue [low exchange] and red [high exchange]). (B) Per-residue root mean square fluctuation (RMSF) of the S protein mapped onto the surface of the S trimer. Deuterium exchange-based dynamics across N-terminal domain (NTD) (C), receptor binding domain (RBD) (D), and the S2 subunit (E). (i) Relative fractional deuterium uptake (RFU) plots of NTD, RBD, and the S2 subunit at 1 min (orange) and 10 min (black) deuterium exchange times, with pepsin digest fragments displayed from N to C-terminus (X-axis). Peptides are grouped into clusters indicated by brackets (X-axis) for ease of display. Individual peptides within each cluster are identifiable from the Supplementary Excel file, which lists clusters and each peptide within each cluster (*Supplementary file 1*: Table S1). (Also see *Figure 2—figure supplement 2*). (ii) Deuterium exchange maps on close-up of the structures of NTD (21–303), RBD (318–552), and the S2 subunit (821–1197). Peptides spanning NTD–RBD interaction sites (166–182, 213–223, 294–303, 318–325, 375–387, and 442–449) showing relatively high deuterium exchange at t = 1 min are highlighted. (iii) The first principal motion and RMSF values of backbone atoms on the NTD, RBD, and the S2 subunit. Residues with high RMSF are labeled. Different domains (fusion peptide [FP], heptad repeat 1 [HR1], central helix [CH], connector domain [CD], heptad repeat 1 [HR2]) showing domain-specific RFU changes are labeled. RFU values are tabulated in *Figure 2—source data 1*.

The online version of this article includes the following source data and figure supplement(s) for figure 2:

**Source data 1.** Relative fractional deuterium uptake values for spike (S) protein peptides at indicated labeling times.

**Figure supplement 1.** Homogeneity of spike (S) protein.

**Figure supplement 2.** Primary sequence coverage map of pepsin proteolyzed peptides of spike S (1–1208).

**Figure supplement 3.** Hydrogen/deuterium exchange mass spectrometry for free spike (S) protein.

**Figure supplement 4.** Dynamics of the spike (S) protein trimer from all-atom molecular dynamics (MD) simulation.

**Figure supplement 5.** Structural stability of full-length spike (S) protein model from all-atom simulations.

The NTD of the S protein showed low overall RFU (~0.2), consistent with its well-structured arrangement of β-sheets connected by loops (*Figures 1B* and *2C*). Importantly, certain regions showed significantly higher deuterium exchange (~0.4), of which two loci (136–143, 243–265) span the dynamic interdomain interactions with the RBD. This is supported by the high per-residue root mean square fluctuations (RMSFs) and large principal motions observed for residues 249–259 during simulations (*Figure 2C*, *Figure 2—figure supplement 4C*). One locus (291–303) at the C-terminal end of the NTD connecting to the RBD showed high deuterium exchange, indicating high relative

motions of the two domains. The RBD (*Figure 2D*) showed an overall higher deuterium exchange (RFU ~0.35), with the peptides spanning the hinge regions (318–336) showing greatest deuterium exchange (~0.6). Peptides spanning residues 351–375 and 432–452 showed significantly increased deuterium uptake, and these correspond to the NTD interdomain interaction sites. Interestingly, certain loci of the RBD at the ACE2 interface (453–467, 491–510) showed higher intrinsic exchange.

Overall, the S2 subunit showed variable deuterium exchange across the constituent domains (*Figure 2E*, *Figure 2—figure supplement 3*). Interestingly, peptides spanning the region directly C-terminal to the S1/S2 cleavage site showed the greatest deuterium exchange (0.6). Congruently, our MD simulations revealed the unstructured loop housing the S1/S2 cleavage site (residues 677–689) to be highly dynamic (*Figure 2—figure supplement 4*), with RMSFs reaching >1.0 nm. It is important to note that the S1/S2 cleavage site has been abrogated in the construct of the S protein used in this study to block proteolytic processing into S1 and S2 subunits during expression in host cells. We observed lower deuterium exchange (and lower RMSF values) at peptides forming the CH and CD, suggesting their function as the central stable core of prefusion S. In contrast, peptides spanning hinge segments and heptad repeats (HR1 and HR2) showed high exchange and RMSF values, reflecting the S protein's ensemble properties encompassing prefusion, fusion, and postfusion conformations in solution.

## Domain-specific and global effects of ACE2 binding to the RBD

Comparative HDXMS of the S protein and S:ACE2 complex showed large-scale changes in S protein upon ACE2 binding. The RBD forms the main interaction site on S protein for ACE2. We therefore set out to comparatively map HDXMS of ACE2:RBD interface of an isolated MBP fusion construct of the RBD ('RBD$_{isolated}$') (*Figure 3C*, *Figure 3—figure supplement 2—source data 1 Supplementary file 1*: Table S2) with S:ACE2 complex (*Figure 4A, B*). A list of peptides common to RBD$_{isolated}$ and S protein ('RBD$_S$') showed differences in deuterium exchange only at interdomain interfaces within individual monomers and trimer interaction sites in the S protein (*Supplementary file 1*: Table S3). Several RBD$_S$ peptides showed decreased exchange upon complexation with ACE2 (*Figure 3*). These include peptides 340–359, 400–420, 432–452, and 487–502 in the RBD$_S$:ACE2 complex (*Figure 4*). Sites showing deuterium exchange protection are consistent with the RBD:ACE2 interface described by X-ray crystallography (PDB: 6M0J) (*Lan et al., 2020*). Further, HDXMS revealed the core of this interface to be contributed by peptides 340–359, 400–420, 432–452, and 491–510 (*Figure 4A, D*, *Figure 2—figure supplement 3*). Interestingly, loci showing large-magnitude differences in deuterium exchange correlate to certain mutational hotspots (*Wang et al., 2020b*).

A closer examination of the RBD$_{isolated}$:ACE2 interface by HDXMS also revealed decreased exchange in peptides spanning these regions (*Figure 3*). However, the magnitude of deuterium exchange protection was significantly more in RBD$_{isolated}$ than in RBD$_S$, potentially reflecting the higher flexibility in the full-length S trimer relative to free RBD, interfering with ACE2 binding. High-resolution structures of RBD:ACE2 reveal the core of the RBD interface to be formed by amino acids Y449, Y453, N487, Y489, G496, T500, G502, Y505, L455, F456, F486, Q493, Q498, and N501 (*Wang et al., 2020a*). These correspond to peptide 448–501 from S protein and RBD$_{isolated}$ in our HDXMS study.

Cryo-EM studies have shown that each RBD in the trimeric S protein can adopt an open conformation irrespective of other RBDs, indicating an absence of cooperativity between the three RBDs within a trimer (*Ke et al., 2020*). Therefore, we compared the deuterium exchange profiles of RBD$_{isolated}$ with RBD$_S$ and observed differences in dynamics imposed by quaternary contacts (*Figure 3*). Overall, the loci with high and low deuterium exchange profiles were similar when compared between RBD$_{isolated}$ and RBD$_S$, both at the disordered ACE2 receptor binding region and the folded regions at the N- and C-termini. In solution, RBD$_S$ toggles between open and closed conformations, resulting in an average readout of deuterium exchange measurements.

ACE2 binding to RBD$_{isolated}$ and RBD$_S$ resulted in similar effects, where we observed deuterium exchange protection at the peptide regions spanning the known binding interface of RBD. Notably, increased deuterium exchange was observed at the hinge region (*Figure 3D*), indicating allosteric conformational changes, associated with restricting the open and closed states interconversion. Therefore, the destabilization/local unfolding observed at the hinge region as a result of ACE2 binding enables RBD to maintain the open conformation. It therefore seems likely that small molecules

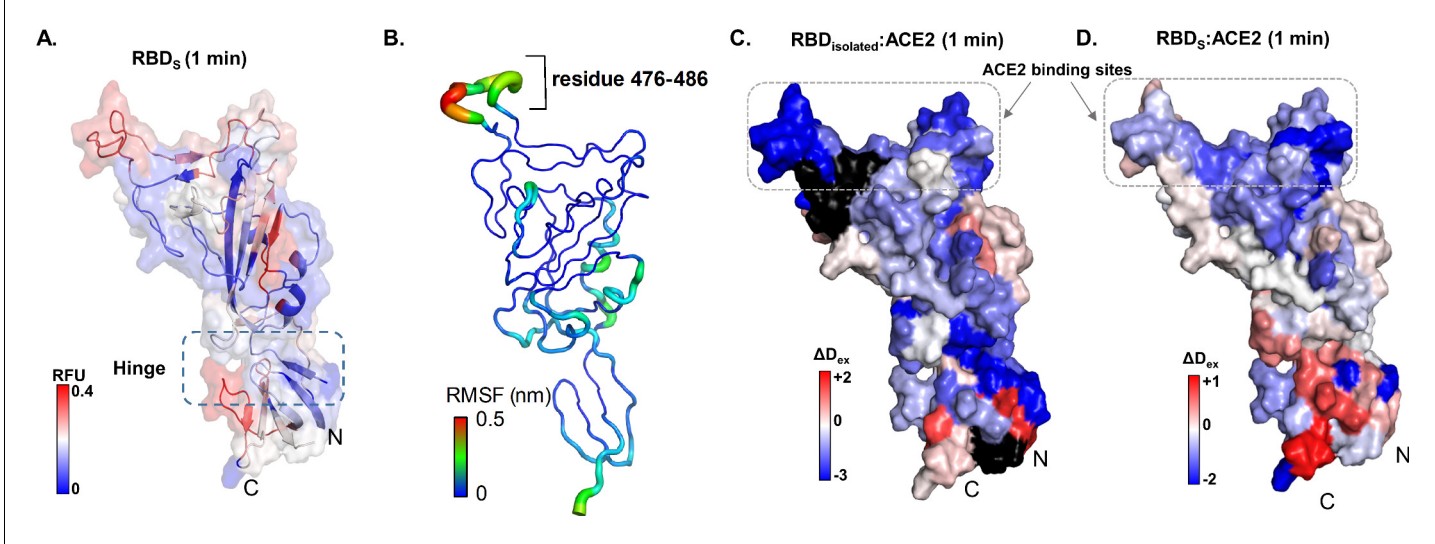

**Figure 3.** Map of receptor binding domain (RBD)$_{isolated}$:angiotensin-converting enzyme 2 (ACE2) interactions. (**A**) Relative fractional deuterium uptake values at t = 1 min for RBD (314–547) of spike (S) protein (RBD$_S$) mapped onto the structure of RBD extracted from S protein model (see ***Supplementary file 1***: Table S2). High and low exchanging regions are represented as shown in key, and regions with no coverage are shown in black. (**B**) The root mean square fluctuation (RMSF) values of backbone atoms on the RBD showing residues with high RMSF (476–486) as per key. Differences in deuterium exchanged between RBD$_{isolated}$:ACE2 complex and free RBD$_{isolated}$ (**C**) and RBD$_S$:ACE with free RBD$_S$ (**D**) at 1 min of deuterium labeling are mapped onto the structure of RBD. Protection from deuterium uptake and increases in exchange are indicated in blue and red, respectively. Regions with no peptide coverage are in black. RFU: relative fractional deuterium uptake.

The online version of this article includes the following source data and figure supplement(s) for figure 3:

**Figure supplement 1.** Homogeneity of receptor binding domain (RBD) isolated sample and validation of angiotensin-converting enzyme 2 (ACE2):RBD complex formation.

**Figure supplement 2.** Primary sequence coverage and deuterium exchange profile of receptor binding domain (RBD)$_{isolated}$.

**Figure supplement 2—source data 1.** Relative fractional deuterium uptake values at different labeling times for pepsin digest fragments of receptor binding domain (RBD)$_{isolated}$.

and biologics targeting the hinge region to lock RBD in the closed state would be of potential high therapeutic value.

## ACE2 binding to RBD is allosterically propagated to the S1/S2 cleavage site and HR

Unexpectedly, ACE2 binding at the RBD induced large-scale changes in deuterium exchange in distal regions of the S protein. Some of the peptides in the stalk of S protein showed decreased exchange in the S:ACE2 complex (***Figure 4C,D***). This indicates that ACE2 receptor interactions stabilized the hinge dynamics in the S protein. Decreased exchange was also seen in the distal sites in the S2 subunit, localized at the FP locus and CH. Interestingly, increased exchange was seen in multiple peptides flanking the S1/S2 cleavage site, HR1 domain, and critically at the S1/S2 cleavage site (***Figure 4D***). Even though the protease cleavage site is abrogated in the construct used in this study, we still observed increased dynamics as inferred by the higher relative deuterium exchange at the S1/S2 locus. Furthermore, this region exhibited high RMSF values during simulations (***Figure 2—figure supplement 4B***). These results clearly indicate that ACE2 binding induces allosteric enhancement of dynamics at this locus, providing mechanistic insights into the conformational switch from the prefusion to fusogenic intermediate. Differences in deuterium exchange between free S protein and the S:ACE2 complex show stabilization at the ACE2 interacting site and local destabilization at peptides juxtaposed to the S1/S2 cleavage site and HR1 ( peptides 931–938). This suggests that ACE2 binding allosterically primes HR1 and other high exchanging regions flanking the S1/S2 cleavage site for enhanced furin protease binding and cleavage. Importantly, these results suggest that the S1/S2 cleavage site is a critical hotspot for S protein dynamic transitions for facilitating SARS-

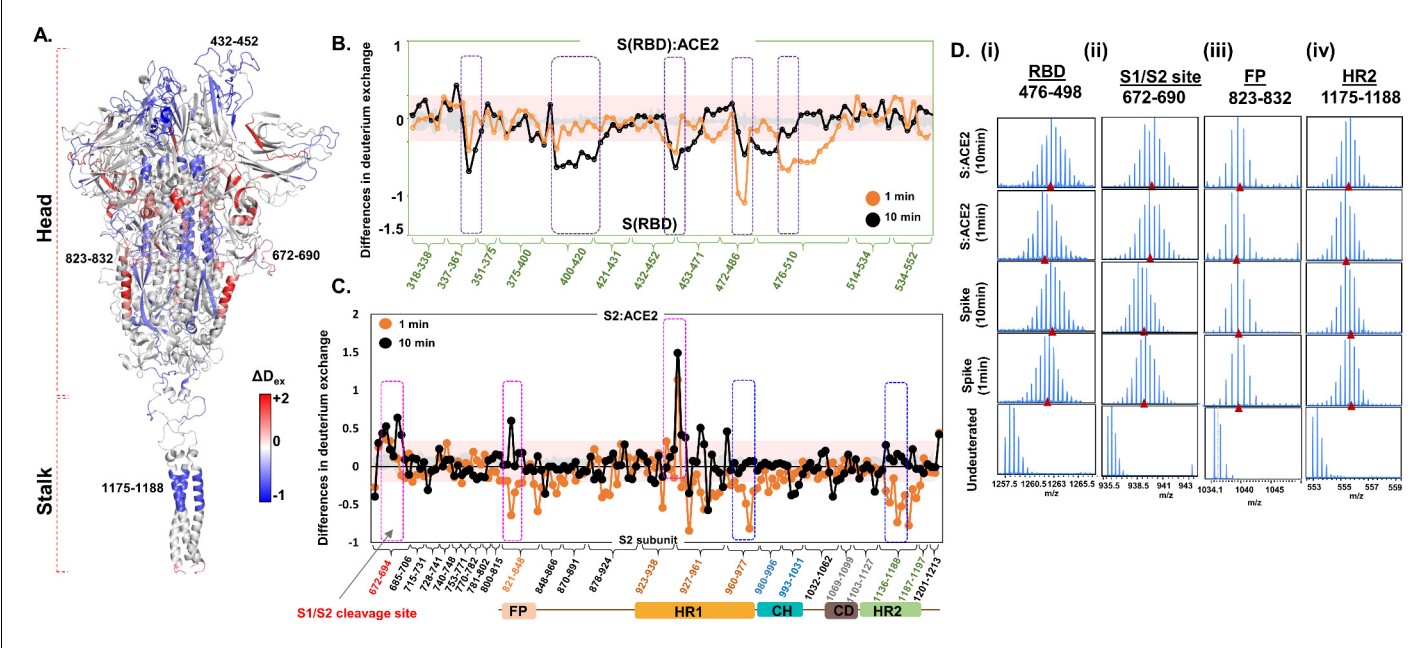

**Figure 4.** Angiotensin-converting enzyme 2 (ACE2) interaction induces large-scale allosteric changes across spike (S) protein. (A) Differences in deuterium exchange ($\Delta D_{ex}$) (t = 1 min) in S protein upon binding ACE2 showing decreased (blue) and increased (red) deuterium exchange, mapped onto the structure of S protein. Deuterium exchange differences (X-axis) for peptides from (B) receptor binding domain (RBD)$_S$ and S2 subunit (C). Peptides are grouped into clusters indicated by brackets (X-axis) for ease of display. Individual peptides within each cluster are identifiable from the source data (*Figure 4—source data 1*). Difference cutoff ±0.3 D (*Houde et al., 2011*) is the deuterium exchange significance threshold indicated by pink shaded box with standard error values in gray. Positive differences (>0.3 D) denote increased deuterium exchange, and negative differences (<−0.3 D) denote decreased deuterium exchange in S protein bound to ACE2. (B) Peptides spanning residues interacting with ACE2 are in purple. (C) Peptides spanning S1/S2 cleavage site, fusion peptide (FP) and heptad repeat 1 (HR1) are highlighted in pink boxes, while peptides spanning central helix and heptad repeat 2 (HR2) are in blue. (D) Stacked mass spectra with isotopic envelopes after deuterium exchange (t = 1, 10 min) for select peptides from (i) RBD (residues 476–498), (ii) S1/S2 cleavage site (residues 672–690), (iii) FP (residues 823–832), and (iv) HR2 (residues 1175–1188) are shown for the S protein and S:ACE2 complex. Mass spectra of the equivalent undeuterated peptide are shown for reference. The centroid masses are indicated by red arrowheads.

The online version of this article includes the following source data for figure 4:

**Source data 1.** Differences in deuterium exchange values between spike (S):angiotensin-converting enzyme 2 (ACE2) complex and free S protein at indicated labeling times.

CoV-2's entry into the host, and therefore represents a new target for inhibitory therapeutics against the virus.

## Dynamics of RBD:ACE2 and S:ACE2 protein interactions provides insights for viral–host entry

Considering the indispensable role of ACE2 binding in SARS-CoV-2 infection, it is crucial to assess the effects of S protein and RBD binding on ACE2 dynamics (*Figure 5*, *Figure 5—figure supplements 1–3*, *Supplementary file 1*: Table S4). We therefore mapped the corresponding binding sites of RBD, both isolated and within the S protein, onto ACE2. The S:ACE2 complex represents the prefusion pre-cleavage state wherein full-length S protein is bound to the ACE2 receptor (*Figure 1B*, ii), while the RBD$_{isolated}$:ACE2 complex represents the post-furin cleavage product formed by the S1 subunit and ACE2 (*Figure 1B*, iii). Previous studies have shown that 14 key amino acids of RBD interact with ACE2, wherein mutations at six sites resulted in higher binding affinity of SARS-CoV-2 (*Li et al., 2005*). SARS-CoV-2 adopted a different binding mode to ACE2 as a superior strategy for infection compared to SARS-CoV-1. A crystal structure of RBD$_{isolated}$:ACE2 complex has identified 24 key ACE2 residues, spanning across peptides 16–45, 79–83, 325–330, 350–357, and R393 (*Towler et al., 2004*). While most of these residues are conserved in binding to both SARS-CoV-1 and SARS-CoV-2, R393 and residues 325–330 are unique to SARS-CoV-1

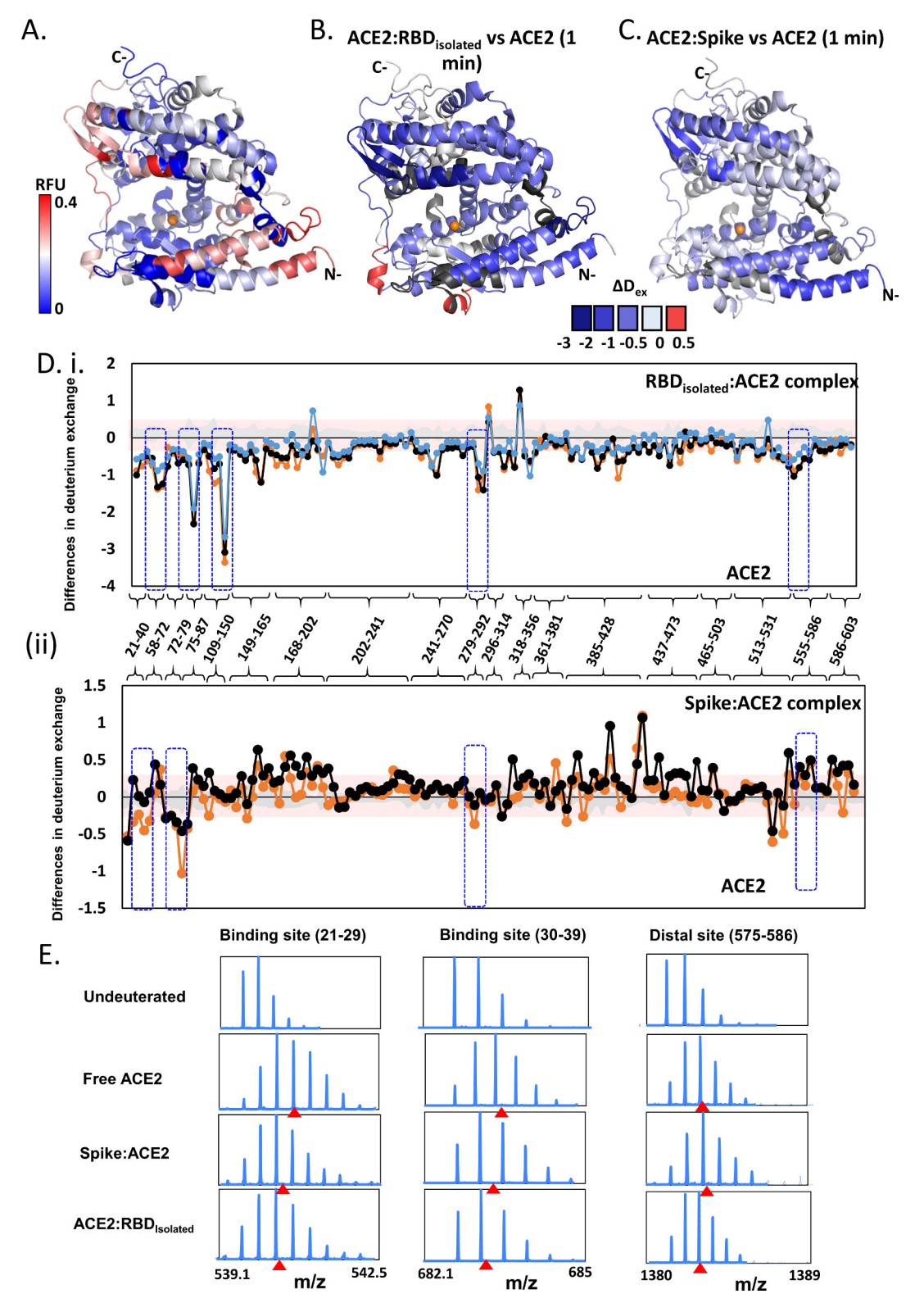

**Figure 5.** Effect of receptor binding domain (RBD)$_{isolated}$ and RBD$_S$ complexes on angiotensin-converting enzyme 2 (ACE2) dynamics. (**A**) Structure of extracellular domain of ACE2 receptor (PDB ID: 1R42) depicting the relative fractional deuterium uptake (RFU) at t = 1 min. (**B**) Differences in deuterium exchange of RBD$_{isolated}$:ACE2 complex and free ACE2 at t = 1 min mapped onto the structure of ACE2, predominantly showing decreased deuterium exchange in ACE2 (shades of blue). (**C**) Heat map of differences in deuterium exchange (t = 1 min) of S:ACE2 complex and free ACE2. (**D**) Plot showing
*Figure 5 continued on next page*

*Figure 5 continued*

differences in deuterium exchange between ACE2 and complexes with RBD$_{isolated}$ (i) and S (ii) at different labeling times. Peptides are grouped into clusters for ease of display and listed in source data (*Figure 5—source data 1*). Cutoff ± 0.3 D is the deuterium exchange significance threshold, indicated by pink shaded box, and standard errors are in gray. Positive differences denote increased deuterium exchange in (i) RBD$_{isolated}$:ACE2 or (ii) S: ACE2 compared to free ACE2, while negative differences denote decreased deuterium exchange. Peptides spanning the sites of interaction with RBD and two distal sites (278–292, 574–585) are highlighted. (E) Stacked mass spectra showing isotopic distribution for select peptides spanning the binding sites (*Ali and Vijayan, 2020*; *Chan et al., 2020*; *Wang et al., 2020a*; *Watanabe et al., 2020*; *Cai et al., 2020*; *Wang et al., 2020b*; *Li et al., 2005*; *Towler et al., 2004*; *Hamuro et al., 2008*; *Hoofnagle et al., 2003*; *Houde et al., 2011*; *Šali and Blundell, 1993*; *Hakansson-McReynolds et al., 2006*; *Dev et al., 2016*; *Eramian et al., 2006*; *Ramachandran et al., 1963*; *Petit et al., 2007*; *van Meer, 1998*; *Krijnse-Locker et al., 1994*) and a distal allosteric site (575–586) for ACE2, S:ACE2, and RBD$_{isolated}$:ACE2 are shown at 1 min deuterium labeling time. Centroids indicated by red arrowheads.

The online version of this article includes the following source data and figure supplement(s) for figure 5:

**Source data 1.** Differences in deuterium exchange between receptor binding domain (RBD)$_{isolated}$:angiotensin-converting enzyme 2 (ACE2) and spike (S):ACE2 complexes with free ACE2 at indicated labeling times.
**Figure supplement 1.** Homogeneity of angiotensin-converting enzyme 2 (ACE2) protein samples and validation of ACE2:receptor binding domain (RBD) complex formation.
**Figure supplement 2.** Pepsin digest map and sequence coverage of angiotensin-converting enzyme 2 (ACE2).
**Figure supplement 2—source data 1.** Relative fractional deuterium uptake values for pepsin digest fragments of angiotensin-converting enzyme 2 (ACE2) at indicated labeling times.
**Figure supplement 2—source data 2.** Differences in deuterium exchange between spike (S):angiotensin-converting enzyme 2 (ACE2) complex minus receptor binding domain (RBD)$_{isolated}$:ACE2 complex for peptides of ACE2 at indicated labeling times.
**Figure supplement 3.** Deuterium uptake profile for angiotensin-converting enzyme 2 (ACE2) receptor and all-atom molecular dynamics (MD) simulation of the ACE2-B$^0$AT1 complex.

interaction (*Wang et al., 2020b*). Interestingly, we observed increased deuterium exchange at these residues in the S:ACE2 complex compared to ACE2 alone (*Figure 5C*). Identifying the intrinsic dynamics and allosteric changes upon binding could guide development of therapeutic antibodies and small molecule drugs.

Simulations of the ACE2 dimer complexed with the B$^0$AT1 amino acid transporter (PDB: 6M1D) (*Yan et al., 2020*) in a model epithelial membrane revealed a large motion of the peptidase domain, which recognizes the S protein RBD, with respect to the transmembrane and juxtamembrane domains (*Figure 5—figure supplement 3*). This large motion is reminiscent of the flexible tilting displayed by the S protein ECD itself, suggesting that both S protein and ACE2 have adaptable hinges that allow for orientational freedom of the domains involved in recognition. To understand how S protein binding affects ACE2 dynamics, we performed HDXMS experiments of monomeric ACE2 alone, S:ACE2 and RBD:ACE2 complexes (*Figure 5*, *Figure 5—figure supplement 2*) and mapped the deuterium exchange values onto a deletion construct of ACE2 (PDB: 1R42) (*Towler et al., 2004*; *Figure 5*, *Figure 5—figure supplement 2*). We observed a reduction in deuterium exchange across both RBD$_{isolated}$:ACE2 and larger S:ACE2 complexes compared to free ACE2 (Figure S8B and S8C). Differences in deuterium exchange between RBD$_{isolated}$:ACE2 complex and free ACE2 showed that RBD binding stabilizes ACE2 globally, specifically large differences at the binding site (peptides 21–29, 30–39, and 75–92), and also at distal regions (peptides 121–146, 278–292, and 575–586) from the RBD binding site of ACE2 (*Figure 5E*). Cryo-EM studies have shown that a dimeric full-length ACE2 receptor can stably bind to one trimer of the S protein (*Yan et al., 2020*).

## Conclusions

Here, a combination of HDXMS and MD simulations provides a close-up of S protein dynamics in the prefusion, ACE2-bound, and other associated conformations. Our results reveal the energetics of the S:ACE2 complex interface. ACE2 binding to the isolated RBD and S protein alike leads to binding and stabilization. Interestingly, ACE2 binding to the RBD induces global conformational changes across the entire S trimer. Importantly, the stalk region undergoes dampening of conformational motions while showing increased deuterium exchange at the proteolytic processing sites. This study may help in explaining how mutations in emerging strains in the ongoing COVID-19 outbreak might alter dynamics and allostery of ACE2 binding and offer a mechanistic basis for altered infectivities observed in emerging strains. Sites on S protein showing altered deuterium exchange describe

allosteric propagation of ACE2 binding and represent novel cryptic targets for therapeutic small molecule inhibitor/antibody discovery.

# Materials and methods

## Key resources table

| Reagent type (species) or resource | Designation | Source or reference | Identifiers | Additional information |
|---|---|---|---|---|
| Gene (*SARS-CoV-2*) | pTT5 expression vector | GenBank | QHD43416.1 | For recombinant S protein |
| Gene (*ACE2*) | pHL-sec expression vector | GenBank | AB046569.1 | For recombinant ACE2 protein |
| Cell line (*Homo sapiens*) | Human embryonic kidney (HEK293-6E) | NRC, Canada | RRID:CVCL_HF20 | |
| Cell line (*Homo sapiens*) | Expi293F | Thermo Fisher Scientific | RRID:CVCL_D615 | |
| Antibody | Anti-human IgG Fc HRHorseradish Peroxidase (HRP) (goat polyclonal) | Thermo Scientific | RRID:AB_2536544 | WB (1:5000) |
| Recombinant DNA reagent | pHLmMBP-10 (plasmid) | Addgene, USA | RRID:Addgene_72348 | For recombinant RBD protein |
| Recombinant DNA reagent | pTT5 expression vector (plasmid) | Addgene, USA | RRID:Addgene_52367 | |
| Recombinant DNA reagent | pHL-sec expression vector (plasmid) | Addgene, USA | RRID:Addgene_99845 | recombinant DNA reagent |
| Chemical compound, drug | 3,3′,5,5′-Tetramethylbenzidine | Sigma Aldrich | RRID:AB_2336758 | |
| Chemical compound, drug | Deuterium oxide (chemical) | Cambridge Isotope Laboratories | CAS# 7789-20-0 | Deuterium exchange experiments |
| Software, algorithm | DynamX | Waters Corporation (Milford, MA) | | Version 3.0 |
| Software, algorithm | ProteinLynx Global Server (PLGS) | Waters Corporation (Milford, MA) | | Version 3.0.1 |
| Software, algorithm | GraphPad Prism software | GraphPad Prism (https://graphpad.com) | RRID:SCR_002798 | Version 5.0.0 |
| Software, algorithm | Modeller | 1989–2020 Andrej Sali | RRID:SCR_008395 | Version 9.21 |
| Software, algorithm | Visual molecular dynamics | University of Illinois at Urbana-Champaign | RRID:SCR_001820 | Version 1.9.3 |

## Materials

Mass spectrometry grade acetonitrile, formic acid, and water were from Fisher Scientific (Waltham, MA); deuterium oxide was from Cambridge Isotope Laboratories (Tewksbury, MA). All reagents and chemicals were research grade or higher and obtained from Merck-Sigma-Aldrich (St. Louis, MO).

## Methods

### Transient expression and purification of recombinant SARS-CoV-2 spike, RBD, and ACE2 receptor

A near-full-length S protein of SARS-CoV-2 (1–1208; Wuhan-Hu-1; GenBank: QHD43416.1), excluding TD and CT, was codon optimized for mammalian cell expression and cloned into pTT5 expression vector with a twin strep tag at the C-terminus (Twist Biosciences, Singapore). Mutations were introduced into this construct at two sites: (i) RRAR motif at the S1/S2 cleavage site (682–685) was substituted by GSAS and (ii) KV motif (986–987) was substituted with two prolines. A gene encoding SARS-CoV-2-RBD (319–591 of SARS-CoV-2 spike) (BioBasic, Singapore) was cloned into the

expression vector pHLmMBP-10 as a fusion protein with N-terminal mMBP and C-terminal hexahistidine tags. A gene encoding human ACE2 (residues 21–597) fused to a C-terminal Fc-tag (BioBasic, Singapore) was cloned into vector pHL-sec between the signal peptide and C-terminal 6xHis tag. S (1–1208) was expressed in HEK293-6E using polyethylenimine as the transfection reagent while the isolated RBD ('RBD$_{isolated}$') and ACE2 constructs were expressed in Expi293F using the Expi293 System. Culture supernatant was harvested on day 7 for HEK293-6E expression and day 5 for Expi293F expression. S protein was affinity purified using Strep-TactinXT column (IBA), RBD protein was affinity purified using cOmplete His-Tag Purification column (Merck, Darmstadt, Germany), and ACE2 receptor was affinity purified using HiTrap MabSelect SuRe column (GE Healthcare, Chicago, IL, USA). Purified proteins were concentrated and buffer exchanged into phosphate buffered saline (PBS) using VivaSpin, and the purity was assessed by denaturing polyacrylamide gel electrophoresis (*Figure 2—figure supplement 1A*, *Figure 3—figure supplement 1A*, and *Figure 5—figure supplement 1A*). Cell lines obtained commercially are listed in key resources table and were tested for contamination by *Mycoplasma* species.

## Characterization of RBD:ACE2 receptor binding

Interactions between recombinant purified MBP-RBD and ACE2 receptor (*Figure 3—figure supplement 1A* and *Figure 5—figure supplement 1A*) were confirmed by enzyme-linked immunosorbent assay. To test binding activity of ACE2, 96-well maxisorp plates were coated with 100 µL of 27.2 nM MBP-RBD diluted in PBS at 4˚C for 16 hr and blocked with 350 µL of 4% skimmed milk in PBST (0.05% Tween 20 in PBS) at room temperature for 1.5 hr. This was followed by 1 hr incubation with ACE2 (100 µL) at varying concentrations and detection with 100 µL of goat-anti-human IgG Fc HRP diluted at 1:5000 in 2% skimmed milk in PBST for 1 hr. Plates were washed three times in PBST after each incubation step above. After 5 min incubation with 100 µL of 3,3′,5,5′-tetramethylbenzidine, reaction was stopped with 100 µL of 1 M $H_2SO_4$ and absorbance at 450 nm ($A_{450}$) was recorded. A similar protocol was adopted for the quality testing of MBP-RBD – it was coated at variable concentrations in PBS at 4˚C for 16 hr and blocked at room temperature for 1.5 hr. This was followed by 1 hr incubation with 10.4 nM ACE2 (100 µL) diluted in blocking buffer. Detection, plate washing, and color development steps were performed in the same manner as described above. Data represents an average of three replicates, along with their error bars and plotted using GraphPad Prism 5 (San Diego, CA).

## Deuterium exchange

S protein (8 µM), ACE2 (52 µM), and RBD (67 µM) solubilized in PBS (pH 7.4) were incubated at 37˚C in PBS buffer reconstituted in $D_2O$ (99.90%), resulting in a final $D_2O$ concentration of 90%. S:ACE2 and RBD$_{isolated}$:ACE2 complexes ($K_D$ of ~15 and ~150 nM, respectively) (*Wrapp et al., 2020*) were pre-incubated at 37˚C for 30 min in a 1:1 molar ratio to achieve >90% binding prior to each hydrogen–deuterium exchange reaction. Deuterium labeling was performed for 1, 10, and 100 min for isolated construct of RBD, free ACE2, and RBD$_{isolated}$:ACE2 complex. For isolated S protein and S:ACE2 complex, 1 and 10 min labeling timescales were used. Pre-chilled quench solution 1.5 M GnHCl and 0.25 M Tris(2-carboxyethyl) phosphine-hydrochloride was added to deuterium exchange reaction mixture to lower the pH$_{read}$ to ~2.5 and lower the temperature to ~4˚C. Next, the quenched reaction was incubated at 4˚C on ice for 1 min followed by online pepsin digestion.

## Mass spectrometry and peptide identification

Approximately 100 pmol quenched samples were injected onto chilled nanoUPLC HDX sample manager (Waters, Milford, MA). The injected samples were subjected to online digestion using immobilized Enzymate BEH pepsin column (2.1 × 30 mm) (Waters, Milford, MA) in 0.1% aqueous formic acid at 100 µL/min. Simultaneously, the proteolyzed peptides were trapped in a 2.1 × 5 mm C18 trap (ACQUITY BEH C18 VanGuard Pre-column, 1.7 µm, Waters, Milford, MA). Following pepsin digestion, the proteolyzed peptides were eluted using acetonitrile gradient of 8–40% in 0.1% formic acid at a flow rate of 40 µL/min into reverse phase column (ACQUITY UPLC BEH C18 Column, 1.0 × 100 mm, 1.7 µM, Waters, Milford, MA) pumped by nanoACQUITY Binary Solvent Manager (Waters, Milford, MA). Electrospray ionization mode was used to ionize peptides sprayed onto SYNAPT G2-Si mass spectrometer (Waters, Milford, MA) acquired in HDMS$^E$ mode. A flow rate of 5 µL/

min was used to continually inject 200 fmol µL$^{-1}$ of [Glu$^1$]-fibrinopeptide B ([Glu$^1$]-Fib) as lockspray reference mass.

For identification of the resolved and eluted peptides, HDMS$^E$ method was used with ion-mobility settings 600 m/s wave velocity and 197 m/s transfer wave velocity. Low collision energies of 4 and 2 V were used for trap and transfer, respectively, while high collision energy was ramped from 20 to 45 V. A constant 25 V cone voltage was used, and mass spectra within 50–2000 Da were acquired for 10 min with mass spectrometer operated in positive ion mode.

Undeuterated protein samples were used to identify sequences from mass spectra data (in HDMS$^E$ mode) using ProteinLynx Global Server (PLGS) v3.0. Peptide identification search was performed against a separate sequence database of each protein sequence, along with its respective affinity purification tag sequences. PLGS search parameters selected for peptide sequence identification were (i) no specific protease and (ii) variable N-linked glycosylation modification. Additional cut-off filters applied included (i) minimum intensity = 2500, (ii) minimum products per amino acids = 0.2, and (iii) a precursor ion mass tolerance of <10 ppm in DynamX v.3.0 (Waters, Milford, MA) and confirmed for pepsin cleavage specificity as described (*Hamuro et al., 2008*). Peptides independently identified under the specified condition and present in at least in two out of three undeuterated sample replicates were retained for HDXMS analysis. S protein contains 22 variable glycosylation sites (*Watanabe et al., 2020*), out of which we identified peptides spanning 12 glycosylation sites in our sample (*Figure 2—figure supplement 2*). However, none of these peptides showed glycosylation. For ACE2, we obtained peptides overlapping four glycosylation sites (*Figure 5—figure supplement 3*).

RFU is the ratio of number of deuterons exchanged to the total number of exchangeable amides of the peptide. Centroid masses of undeuterated reference spectra were subtracted from equivalent spectra of deuterium exchanged peptides to calculate the average deuterons exchanged for each peptide. Deuterium exchange plots, relative deuterium exchange, and difference plots were generated by DynamX v.3.0. The N-terminus and all prolines in each peptide were excluded for estimation of exchangeable amides per peptide (*Hoofnagle et al., 2003*). Deuterium exchange experiments for two biological replicates and technical triplicates of S protein and the S:ACE2 complex were carried out. Average deuterium exchange measurements between the two biological replicates were within ±0.3 Da (*Supplementary file 1*: Table S5, S6) (*Houde et al., 2011*). While deuterium exchange values are not corrected for back exchange, fully deuterated S protein samples were used to measure deuterium back exchange. A list of peptides with back exchange values is shown in *Supplementary file 1*: Table S7. The mass spectrometry proteomics data have been deposited to the ProteomeXchange Consortium via the PRIDE [1] partner repository with the dataset identifier PXD023138.

## Modeling and MD simulations

An integrative model of full-length SARS-CoV-2 S protein was built using Modeller v.9.21 (*Šali and Blundell, 1993*). The cryo-EM structure of prefusion S ECD in the open conformation (PDB: 6VSB) (*Wrapp et al., 2020*) was used as the template for the ECD with missing loops on the NTD and the C-terminus of the ECD modeled based on the cryo-EM structure of S ECD in the closed conformation resolved at a higher resolution (PDB: 6XR8) (*Cai et al., 2020*). The Nuclear Magnetic Resonance (NMR) structure of the SARS S HR2 domain (PDB: 2FXP) (*Hakansson-McReynolds et al., 2006*) was used as the template for the HR2 domain, while the TM domain was modeled using the NMR structure of the HIV-1 gp-41 TM domain (PDB: 5JYN) (*Dev et al., 2016*). Ten models were built and subjected to stereochemical assessment using the discreet optimized protein energy (DOPE) score (*Eramian et al., 2006*) and Ramachandran analysis (*Ramachandran et al., 1963*). The model with the lowest DOPE score and the smallest number of Ramachandran outliers was chosen. Palmitoylation was performed at three cysteine residues (C1236, C1240, and C1243) on the CT domain based on a study showing its importance in SARS S protein function (*Petit et al., 2007*). The S protein model was then embedded into a model membrane representing the endoplasmic reticulum–Golgi intermediate compartment (ERGIC) (*van Meer, 1998*), where coronaviruses are known to assemble in a bud form (*Krijnse-Locker et al., 1994*; *Klumperman et al., 1994*). The ERGIC model membrane was built using CHARMM-GUI Membrane Builder (*Lee et al., 2019*).

All-atom MD simulation was performed for 200 ns using GROMACS (University of Groningen, Netherlands) 2018 (*Abraham et al., 2015*) and the CHARMM36 force field (*Huang and MacKerell, 2013*). The system was solvated with 590,742 TIP3P water molecules and 0.15 M NaCl salt, achieved by adding 3235 Na+ and 2103 Cl⁻ ions. Minimization and equilibration were performed following standard CHARMM-GUI protocols (*Lee et al., 2016*). This includes six steps of equilibration; the first two steps used a 1 fs integration time step for 125 ps, while the last four used 2 fs time step for 250 ps. With each step, the magnitude of positional and dihedral restraints imposed on the protein and lipid molecules was gradually reduced by lowering the force constants from 1000 (step 1) to 0 kJ $mol^{-1}$ $nm^{-2}$ (step 6). Temperature and pressure were maintained at 310 K and one atm, respectively, using the Berendsen thermostat and barostat during equilibration. This was then followed by the production run, whereby the temperature was maintained using the Nosé–Hoover thermostat (*Nosé, 1984*; *Hoover, 1985*) and the pressure was maintained via semi-isotropic coupling to the Parrinello–Rahman barostat (*Parrinello and Rahman, 1981*). Electrostatics were calculated using the smooth particle mesh Ewald method (*Essmann et al., 1995*) with a real space cutoff of 1.2 nm and the van der Waals interactions were truncated at 1.2 nm with force switch smoothing between 1.0 and 1.2 nm. Constraints were applied to covalent bonds with hydrogen atoms using the LINCS algorithm (*Hess et al., 1997*) and a 2 fs integration time step was employed. Snapshots of the trajectory were saved every 100 ps. To assess whether the system was properly equilibrated, we calculated domain-specific root mean square deviations (RMSDs) of the Cα atoms following least-squares fitting (*Figure 2—figure supplement 5*). For all three domains tested (NTD, RBD, and HR2) in all three chains of the S protein, the RMSD reached a plateau after around 50 ns. Additionally, we also calculated RMSF profiles using 20 ns trajectory windows along the simulations. Similarly, the per-residue RMSF values for all three domains converged after the first three windows (60 ns).

For simulations of the ACE2 receptor, the cryo-EM structure of the ACE2-B⁰AT1 complex in the open conformation (PDB: 6M1D) (*Yan et al., 2020*) was used. The ACE2-B⁰AT1 complex was embedded into a model membrane representing the epithelial cell membrane (*Jia et al., 2005*; *Sampaio et al., 2011*). The system was solvated with 314,442 TIP3P water molecules and 0.15 M NaCl salt (1868 Na+ and 1300 Cl⁻ ions). Minimization, equilibration, and production runs were performed using the protocols described above. Principal component analysis and RMSF analyses were performed using GROMACS, and simulations were visualized in VMD (University of Illinois at Urbana-Champaign, USA) (*Humphrey et al., 1996*).

## Acknowledgements

We thank Dr. Lu Gan, Dept. of Biological Sciences, National University of Singapore, Sean Braet, Theresa Buckley and Varun Venkatakrishnan, Dept. of Chemistry, the Pennsylvania State University for helpful discussions. Additionally, we thank reviewers and a reader for their feedback. We thank Protein Production Platform of Nanyang Technological University for their help in making the RBD and ACE2 expression constructs and small-scale protein expression tests. HDXMS experiments were carried out as a fee for service at the Singapore National Laboratory for Mass Spectrometry (SingMass) funded by NRF, Singapore. PVR was supported by research scholarship from National University of Singapore, Singapore. NKT was supported by research grant from Ministry of Education, Singapore awarded to GSA (MOE2017-T2-A40-112). This work was supported by BII of A*STAR. Simulations were performed on the petascale computer cluster ASPIRE-1 at the National Supercomputing Centre of Singapore (NSCC) and the A*STAR Computational Resource Centre (A*CRC).

## Additional information

### Funding

| Funder | Grant reference number | Author |
| --- | --- | --- |
| National Medical Research Council | WBS#R-571-000-081-213 Establishment of assays for drug screening and virus characterization of the newly emerged novel coronavirus (2019-nCoV) | Paul A MacAry |

| | | which is also known as the Wuhan coronavirus |
| --- | --- | --- |
| A*STAR Bioinformatics Institute | | Peter J Bond |
| National University of Singapore | | Palur V Raghuvamsi |
| Ministry of Education - Singapore | MOE2017-T2-A40-112 | Nikhil K Tulsian<br>Ganesh S Anand |

The funders had no role in study design, data collection and interpretation, or the decision to submit the work for publication.

## Author contributions

Palur V Raghuvamsi, Nikhil K Tulsian, Conceptualization, Resources, Data curation, Formal analysis, Visualization, Methodology, Writing - original draft, Writing - review and editing; Firdaus Samsudin, Conceptualization, Data curation, Formal analysis, Visualization, Methodology, Writing - original draft; Xinlei Qian, Kiren Purushotorman, Gu Yue, Mary M Kozma, Resources, Data curation, Methodology; Wong Y Hwa, Resources; Julien Lescar, Methodology; Peter J Bond, Conceptualization, Resources, Data curation, Software, Formal analysis, Supervision, Validation, Investigation, Visualization, Methodology, Writing - original draft, Project administration, Writing - review and editing; Paul A MacAry, Conceptualization, Resources, Data curation, Supervision, Funding acquisition, Validation, Investigation, Project administration, Writing - review and editing; Ganesh S Anand, Conceptualization, Formal analysis, Supervision, Validation, Investigation, Writing - original draft, Project administration, Writing - review and editing

## Author ORCIDs

Palur V Raghuvamsi ⬤ https://orcid.org/0000-0002-0897-6935
Nikhil K Tulsian ⬤ https://orcid.org/0000-0001-6577-6748
Peter J Bond ⬤ https://orcid.org/0000-0003-2900-098X
Ganesh S Anand ⬤ https://orcid.org/0000-0001-8995-3067

## Decision letter and Author response

Decision letter https://doi.org/10.7554/eLife.63646.sa1
Author response https://doi.org/10.7554/eLife.63646.sa2

# Additional files

## Supplementary files

• Supplementary file 1. Table S1. Deuterium exchange at indicated labeling times for S and S:ACE2 complex. Table S2. Deuterium exchange at indicated labeling times for $RBD_{isolated}$ and ACE2 bound $RBD_{isolated}$. Table S3. Comparison of deuterium exchange of peptides common to RBD(isolated) and RBD(Spike) in free and ACE2-bound states at indicated labeling times. Table S4. Deuterium exchange at indicated labeling times for free ACE2 and its complexes with $RBD_{isolated}$ and S protein. Table S5. Comparison of deuterium exchange values for peptides common to biological replicates of S protein at 1 and 10 min labeling. Table S6. Comparison of deuterium exchange of peptides common between the two biological replicates of S and S:ACE2. Table S7. List of peptides of S with deuterium exchange for a fully deuterated state to determine deuterium back-exchange.

• Transparent reporting form

## Data availability

All data generated or analysed during this study are included in the manuscript and supporting files. Source data files have been provided for Figures 2, 3, 4 and 5.

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
