## [Decision Letter]

**Acceptance summary:**

Understanding the mechanism of action of the SARS-Cov-2 virus, which is responsible for the COVID-19 pandemic, at the molecular level is key to developing therapeutics. This work is a timely and interesting exploration of the interaction between the Spike protein on the surface of the SARS-Cov-2 virus and the ACE2 receptor on the surface of human cells. This Spike-ACE2 interaction is an important step in the initial stages of viral infection. The results suggest that the Spike-ACE2 interaction induces extremely long-range allosteric effects on the Spike protein that could help trigger proteolysis of the Spike protein. It is anticipated that the results of this work would have implications for the development of small molecule inhibitors to prevent SARS-Cov-2 infection.

**Decision letter after peer review:**

Thank you for submitting your article "SARS-CoV-2 S protein ACE2 interaction reveals novel allosteric targets" for consideration by *eLife*. Your article has been reviewed by three peer reviewers, including Donald Hamelberg as the Reviewing Editor and Reviewer #1, and the evaluation has been overseen by Olga Boudker as the Senior Editor. The following individual involved in review of your submission has agreed to reveal their identity: Elizabeth Komives (Reviewer #2).

The reviewers have discussed the reviews with one another and the Reviewing Editor has drafted this decision to help you prepare a revised submission.

Summary:

This is a timely and interesting exploration of the interaction between the Spike protein of SARS-CoV-2, the virus responsible for the COVID-19 pandemic, and the ACE2 receptor using hydrogen deuterium exchange mass spectrometry and molecular dynamics simulations. The Spike protein consists of two sub-domains S1 and S2 with the S1 needing to be cleaved-off so the S2 can become the fusion protein responsible for getting the SARS-CoV-2 into the cell. Structures are available but they do not shed light on how the protease furin can access the cleavage site between S1 and S2 in order to begin the process of fusion. The results suggest that the Spike-ACE2 interaction induces extremely long-range allosteric effects on the Spike protein that could trigger proteolysis of the Spike protein. Specifically, when ACE2 binds to the Spike protein, a conformational change occurs near the S1/S2 cleavage site, exposing it and likely making it more susceptible to furin cleavage. The binding also dampens exchange in the stalk region of the Spike protein. The authors refer to these regions as "dynamic hotspots in the pre-fusion state". The results of this work have implications for the development of small molecule inhibitors.

In general, the work is timely, and the results will be of interest to many in the field. The major conclusions of the work are generally supported by the results. Below are suggestions to improve the manuscript and strengthen the conclusions of the work.

Essential revisions:

1) The manuscript appears to have been hastily written, it would benefit from a scientific editor making it more readable. For example, "Average deuterium exchange at these 91 reporter peptides was monitored for comparative deuterium exchange analysis of S protein, ACE2 receptor and S:ACE2 complex, along with a specific ACE2 complex with the isolated RBD." Presumably "reporter peptides" refers to the 321 peptides mentioned two sentences earlier…Why is the ACE2 complex with the isolated RBD qualified as "specific" while none of the others are? Then the article continues with more information about glycosylation…

2) Subsection “Localizing subunit specific dynamics and domain motions of S protein trimer” and Figure 2—figure supplement 2: A bit more should be said about the glycosylation sites. If only non-glycosylated peptides being observed in the pepsin digestion, the coverage map (Figure 2—figure supplement 2), shows expected lack of coverage for only a few sites (17, 122, 149, 165, 234, 282, 709, 1134) whereas many other sites are covered by peptides. Does this indicate that these sites are mostly not glycosylated?

3) Figure 2—figure supplement 3 legend seems to indicate that uptake of each peptide is plotted, whereas uptake per residue should be plotted because overlapping peptides make these data misleading. The peptides are shown in the other relative uptake graphs, but then there is more than one data point per peptide. Can the authors explain a bit more in the legend how they got the data in these figures?

4) Figure 2—figure supplement 4 seems to indicate that the cleavage site is already very dynamic. Can the authors explain this?

5) "… Mapping the relative deuterium exchange across all peptides onto this S protein model showed the greatest deuterium exchange at the stalk region" seems to contradict "The deuterium exchange heat map showed the highest relative exchange in the S2 subunit (Figure 2—figure supplement 3) and helical segments," Please clarify.

6) Figure 2A and B look like the same molecular structure (nice that they are in the same orientation) but the domains are labeled differently. Yet a third domain listing is used in panel E. Comparing panels A and B, it's a little strange that some of the least dynamic spots in the Head/ECD are the highest exchanging, do the authors want to comment on this?

7) If it wouldn't slow things down too much, it would be great if the RFU data were calculated after back exchange correction. Even an imperfect correction (such as a global correction for the back exchange during analysis) would make the data more meaningful.

8) Figure 3C and D look remarkably different considering that they both are reflecting the RBD:ACE2 interaction. Did the authors attempt to find a convergent set of peptides to do this analysis? Perhaps if the binding site were labeled it would help make the differences look less important (overall the top part of the molecule is blue and the bottom more-or-less has some red and if that's all we are supposed to get out of this figure then it is ok).

9) Figure 4. The authors state that the significance cut-off for difference in deuterium exchange is 0.3 D but there is no explanation of how the value were derived.

10) Figure S1: The SDS-PAGE showed around 90 kDa for the molecular weight of RBDisolated, which should be around 25 kDa based on its sequence (318-547). Please check and clarify.

11) It is confusing about the existing forms of the S protein and ACE2 and their binding stoichiometry, regarding the statements such as "we measured dynamics of a trimer of this near-full length S protein…", "we performed HDXMS experiments of monomeric ACE2…", "…were pre-incubated at 37°C for 30 min in a molar ratio of 1:1 to achieve >90% binding…". Please confirm whether the expressed ACE2 is dimeric and S protein is trimeric or not, and their binding stoichiometry is 1:1 or 2:3. Please also provide the concentration and calculation details for ensuring the >90% binding. If only one ACE2 in the ACE dimer and one S protein in the S protein trimer are involved in the binding, how sensitive and accurate could the HDX-MS results reflect the binding, since no HDX difference would be observed for the other ACE2 and other 2 S proteins?

12) Abstract: Other studies (e.g., Hoffman et al., 2020) have shown that ACE2 binding can enhance S1/S2 cleavage by furin and S1/S2 cleavage site could be possible targets for small molecule inhibitor/antibody development. It would be helpful if further evidence could be provided to support that the stalk hinge regions could also be the targets for that.

13) The authors should provide more technical details of the molecular dynamics simulations in the supplementary materials. Could the authors provide more details on the equilibration protocol? Was there any analysis done or metric used to assess whether the system was properly equilibrated? How often were snapshots of the trajectory saved for analysis? How many Na^+^ and Cl^-^ ions were added to achieve 0.15 M of salt concentration? Also, how many water molecules were added? These details are relevant to the non-casual readers.

14) Figure 2—figure supplement 1: Could the authors elaborate on Figure 2—figure supplement 1B in the figure legend? Is (i) measuring the binding of ace2 to the S protein? Is (ii) measuring the binding of RBD to the ace2 protein? The distinction between (i) and (ii) is not made in the figure legend.

---

## [Author Response]

Essential revisions:1) The manuscript appears to have been hastily written, it would benefit from a scientific editor making it more readable. For example, "Average deuterium exchange at these 91 reporter peptides was monitored for comparative deuterium exchange analysis of S protein, ACE2 receptor and S:ACE2 complex, along with a specific ACE2 complex with the isolated RBD." Presumably "reporter peptides" refers to the 321 peptides mentioned two sentences earlier…Why is the ACE2 complex with the isolated RBD qualified as "specific" while none of the others are? Then the article continues with more information about glycosylation…

We accept the reviewer’s suggestions and feedback and have revised and rewritten the manuscript to make it more accessible to the readers and address reviewers’ concerns.

For the example cited above, we have significantly amended the text. The reporter peptides indeed refer to the 321 pepsin fragment peptides (317 in the revised manuscript) used in the HDXMS analysis. We have made it much clearer in the revised manuscript for a broader readership new to HDXMS (subsection “Subunit specific dynamics and domain motions of S protein trimer”). We have amended and removed any qualifier denoting RBD:ACE2 complex as a “specific” complex. We thank the reviewer for their feedback.

2) Subsection “Localizing subunit specific dynamics and domain motions of S protein trimer” and Figure 2—figure supplement 2: A bit more should be said about the glycosylation sites. If only non-glycosylated peptides being observed in the pepsin digestion, the coverage map (Figure 2—figure supplement 2), shows expected lack of coverage for only a few sites (17, 122, 149, 165, 234, 282, 709, 1134) whereas many other sites are covered by peptides. Does this indicate that these sites are mostly not glycosylated?

We thank the reviewers for this query. Spike protein has been predicted to be glycosylated at 22 sites, 20 of which are predicted to be N-linked glycosylations. Spectra with good signal to noise ratios were observed only for 12 glycosylation sites (61, 74, 331, 343, 603, 656, 717, 801 1074, 1098, 1173, 1194) and included in this study. Peptides spanning other 8 (16, 122, 149, 165, 234, 282, 616 709) glycosylation sites were of poor quality and not considered. As pointed out by the reviewers, a peptide identification search with glycosylation filter did not show glycosylation at the 12 analyzed peptides. This is amended in the subsection “Subunit specific dynamics and domain motions of S protein trimer”.

3) Figure 2—figure supplement 33 legend seems to indicate that uptake of each peptide is plotted, whereas uptake per residue should be plotted because overlapping peptides make these data misleading.

We thank the reviewer for pointing this out and have amended the figure legend for Figure 2—figure supplement 3 to explain this better. This figure now shows relative fractional uptake of deuterium (RFU), not absolute deuterium uptake as erroneously reported in our first submission. We have divided peptides (denoted by a dot) into clusters, indicated by brace brackets on the X-axis, for ease of display. Individual peptides within each cluster are identifiable from their source data and Supplementary file 1 which list each cluster and each peptide within that cluster.

We acknowledge the reviewer’s point that the distribution of pepsin fragments as generated in a modified mirror plot by DynamX appear uniform but are in fact a function of propensity of pepsin cleavage, which is in turn a function of protein sequence. Consequently, some clusters have more constituent peptides, some of which may be overlapping and are nonuniform. Not all peptides within each cluster are overlapping and therefore the plot describes deuterium exchange per peptide. We have amended legends for respective figures.

The peptides are shown in the other relative uptake graphs, but then there is more than one data point per peptide. Can the authors explain a bit more in the legend how they got the data in these figures?

We thank the reviewers for their comment and acknowledge lack of clarity in our figure legends surrounding our grouping peptides into clusters. Each point denotes a single peptide but due to the lack of space and for ease of display, we have grouped peptides into clusters. We have now amended this for each figure legend in the main text.

4) Figure 2—figure supplement 4 seems to indicate that the cleavage site is already very dynamic. Can the authors explain this?

In free S protein HDXMS and MD simulations, a single S1/S2 cleavage site spanning peptide 672690 (Figure 4D) includes a loop region with high intrinsic deuterium exchange and high simulated RMSF values (Figure 2—figure supplement 4B). Importantly, upon ACE2 binding, this peptide showed even greater exchange in the S:ACE2 complex. This forms the primary allosteric response site which we propose to enhance furin proteolytic processing of the S:ACE2 complex. We have included an explanation in the subsection “ACE2 binding to RBD is allosterically propagated to the S1/S2 cleavage site and Heptad Repeat”.

5) "… Mapping the relative deuterium exchange across all peptides onto this S protein model showed the greatest deuterium exchange at the stalk region" seems to contradict "The deuterium exchange heat map showed the highest relative exchange in the S2 subunit (Figure 2—figure supplement 3) and helical segments," Please clarify.

We seek to clarify that the stalk region spanning (central helix and heptad repeat) is part of the S2 subunit showing the greatest deuterium exchange. Notably, within the S2 subunit, domain-specific deuterium exchange was observed, of which the stalk region (central helix and heptad repeats) showed greatest magnitude deuterium exchange. In Figure 2—figure supplement 3, average RFU values (indicated by solid red line) of the S1 and S2 subunits are represented in separate plots. These highlight that the S2 subunit shows higher overall deuterium exchange and by extension dynamics relative to the S1 subunit.

6) Figure 2A and B look like the same molecular structure (nice that they are in the same orientation) but the domains are labeled differently. Yet a third domain listing is used in panel E. Comparing panels A and B, it's a little strange that some of the least dynamic spots in the Head/ECD are the highest exchanging, do the authors want to comment on this?

We thank the reviewer for this insight. We have amended the figure in the revised manuscript. Panel E lists the subdomains of the S2 subunit, for which we have added the residue numbers for the peptides. We would like to clarify that 2A and 2B only show exchange and RMSF values of surface exposed peptides and not the interior structures due to surface representation.

7) If it wouldn't slow things down too much, it would be great if the RFU data were calculated after back exchange correction. Even an imperfect correction (such as a global correction for the back exchange during analysis) would make the data more meaningful.

We appreciate the reviewers’ suggestion to include details of deuterium back-exchange. We have carried out deuterium back-exchange control experiments with fully deuterated S protein (subsection “Mass Spectrometry and peptide identification”). A list of peptides with their back exchange values is provided in Table S7 in Supplementary file 1. Back exchange showed a range between 14.7 to 51%. However the median back exchange for multiple peptides is ~ 34%. This does not change the deuterium exchange heat map significantly. The stalk and S1/S2 proteolysis sites continue to show highest exchange. We have therefore not factored this and applied deuterium exchange corrections. We believe that upon back exchange correction, the differences in deuterium exchange values would be more pronounced. We welcome the reviewer’s suggestion to show readers our back exchange for different peptides.

We have included biological replicate measurements of HDXMS (S protein) and report the new data in Tables S5 and S6 in Supplementary file 1. 4 peptides that showed lower signal to noise were excluded from the final list of 317 peptides. This did not change the sequence coverage in our study. Differences in deuterium exchange values between the two biological replicates are within the low standard errors for measurements (± 0.3 Da).

8) Figure 3C and D look remarkably different considering that they both are reflecting the RBD:ACE2 interaction. Did the authors attempt to find a convergent set of peptides to do this analysis? Perhaps if the binding site were labeled it would help make the differences look less important (overall the top part of the molecule is blue and the bottom more-or-less has some red and if that's all we are supposed to get out of this figure then it is ok).

We thank the reviewers for this suggestion. Figure 3C shows the effect of ACE2 binding on the isolated RBD, while Figure 3D is from the RBD from the larger S protein construct. These were used for comparison. We were also interested in comparing ACE2 interactions of a monomer (isolated RBD) with that of a trimer (full-length S protein). Importantly, we found no differences in the interface mapped by HDXMS for these two constructs. Our results suggest that trimeric S protein bound ACE2 marginally less tightly than isolated RBD. This might be due to the intrinsic dynamics of the S protein trimer interfering with ACE2 binding.

We thank the reviewer for their suggestion. We include a table comparing the common peptides between RBD_isolated_ and RBD of S protein (Table S3 in supplementary file 1), which reveal this marginal difference between RBD and S protein. We have included a sentence to describe this in the subsection “Domain-specific and global effects of ACE2 binding to the RBD”.

9) Figure 4. The authors state that the significance cut-off for difference in deuterium exchange is 0.3 D but there is no explanation of how the value were derived.

The significance cut-off for differences in deuterium exchange were experimentally determined from standard deviations of the average deuterium uptake values from technical and biological replicates, which were obtained from DynamX software. All peptides showed a standard deviation within ± 0.3 D. This was the basis for this value to represent the significance threshold for measurements and is consistent with findings from a previous study. (Houde, Berkowitz and Engen. 2011). We have added a sentence to describe the basis for the deuterium exchange significance cutoff to the Figure 4 legend.

10) Figure 2—figure supplement 1: The SDS-PAGE showed around 90 kDa for the molecular weight of RBDisolated, which should be around 25 kDa based on its sequence (318-547). Please check and clarify.

We thank the reviewer for their query. As described in the Materials and methods, we used a MBP fusion construct of RBD in this study. The total molecular weight for our fusion protein adds up to 90 kDa. Figure 2—figure supplement 1 legend has been amended to describe this.

11) It is confusing about the existing forms of the S protein and ACE2 and their binding stoichiometry, regarding the statements such as "we measured dynamics of a trimer of this near-full length S protein…", "we performed HDXMS experiments of monomeric ACE2…", "……were pre-incubated at 37°C for 30 min in a molar ratio of 1:1 to achieve >90% binding……".

We thank the reviewer. We have included size exclusion chromatography (SEC) profiles of MBPRBD, Fc-ACE2, S protein in Figure 2—figure supplement 1, Figure 3—figure supplement 1 and Figure 5—figure supplement 1 respectively, which clearly show their elution profiles as monomers, dimers and trimers respectively. Multiple Cryo-EM studies have also confirmed that this Spike double mutant construct used in the current study is trimeric, while Fc tagged-ACE2 is dimeric in solution (Fc forms dimers).

Please confirm whether the expressed ACE2 is dimeric and S protein is trimeric or not, and their binding stoichiometry is 1:1 or 2:3. Please also provide the concentration and calculation details for ensuring the >90% binding. If only one ACE2 in the ACE dimer and one S protein in the S protein trimer are involved in the binding, how sensitive and accurate could the HDX-MS results reflect the binding, since no HDX difference would be observed for the other ACE2 and other 2 S proteins?

In the Materials and methods section, stoichiometries and calculations of the protein concentrations used have been provided. Earlier studies have reported ACE2 binding affinities of 15 nM and 150 nM to S protein and RBD. We used 1:1 ratio of RBD_isolated_:ACE2 in their monomeric forms. We determined that under our experimental conditions with saturating concentrations of RBD isolated and ACE2, nearly all RBD and ACE2 (>99% bound as per the program % bound; University of California San Diego), indicative of their complexation without any predominant free forms in solution.

For S:ACE2 complex, the open state of RBD alone binds to ACE2. Benton et al. (Nature, 2020, https://doi.org/10.1038/s41586-020-2772-0) have shown that RBD can exist in combination of open and closed state in solution i.e. all closed, one open-, two closed or two open, one closed or three open states making it difficult to predict the exact binding stoichiometries to ACE2. However, we estimate complete saturation binding of ACE2 with S protein at concentrations used in our HDXMS experiments (550nM ACE2:550nM S protein) during deuterium exchange conditions. In our HDXMS reaction mixture we have used 1:1 ratios of S protein (monomer): Fc-ACE2 (monomer) and equilibrated for 30 min prior to performing the deuterium labeling to obtain saturated binding.

We observed no bimodal deuterium exchange kinetics for any S protein or ACE2 peptides which would have been observed for substoichiometric complexation of S with ACE2. Deuterium exchange protection was also only marginally lower for S:ACE2 compared to RBD_isolated_:ACE2, saturation binding under HDXMS experimental conditions.

12) Abstract: Other studies (e.g., Hoffman et al., 2020) have shown that ACE2 binding can enhance S1/S2 cleavage by furin and S1/S2 cleavage site could be possible targets for small molecule inhibitor/antibody development. It would be helpful if further evidence could be provided to support that the stalk hinge regions could also be the targets for that.

We thank the reviewer for this suggestion. The S1/S2 cleavage site for host furin proteases is essential for the S protein to dissociate into individual S1 and S2 subunits for the host-entry process. We envisage this represents a primary hotspot for targeting small molecule inhibitors against SARS-CoV-2.

The stalk provides flexibility to the S protein for binding to ACE2. However, parts of the stalk regions showed contrasting decreased deuterium exchange upon ACE2 binding. Tomogram studies of SARS-CoV-2 virus have shown that S protein binds to ACE2 even at various angles/conformations of the stalk. We don’t have evidence for the efficacy of targeting the stalk hinge regions for better neutralization but rank the S1/S2 cleavage site better as a drug target.

13) The authors should provide more technical details of the molecular dynamics simulations in the supplementary materials. Could the authors provide more details on the equilibration protocol? Was there any analysis done or metric used to assess whether the system was properly equilibrated? How often were snapshots of the trajectory saved for analysis? How many Na^+^ and Cl^-^ ions were added to achieve 0.15 M of salt concentration? Also, how many water molecules were added? These details are relevant to the non-casual readers.

We thank the reviewer for their query. We have now updated our simulations, using an improved S protein model in which missing loops located in the NTD and the C-terminus of the ECD have been modelled based on a higher resolution structure (PDB: 6XR8). The new model nevertheless exhibited comparable dynamics to our previous model.

As requested by the reviewer, we have now added more technical details of the molecular dynamics simulations in the Materials and methods section, including the equilibration protocol, the frequency of saving frames from trajectories, the number of water molecules as well as Na^+^ and Cl^-^ ions to the systems. To assess proper system equilibration, we calculated RMSD curves for subdomains of the S protein and found that they converged after around 50 ns. Likewise, we made similar assessments on the basis of calculation of block-wise RMSF profiles. This new data has been added to Figure 2—figure supplement 5.

14) Figure 2—figure supplement 1: Could the authors elaborate on Figure 2—figure supplement 1B in the figure legend? Is (i) measuring the binding of ace2 to the S protein? Is (ii) measuring the binding of RBD to the ace2 protein? The distinction between (i) and (ii) is not made in the figure legend.

We thank the reviewer for this point. We have provided experimental details of the ELISA performed in the Materials and methods section of the revised manuscript.